# Increasing Stress to Induce Apoptosis in Pancreatic Cancer via the Unfolded Protein Response (UPR)

**DOI:** 10.3390/ijms24010577

**Published:** 2022-12-29

**Authors:** Gehan Botrus, Richard M. Miller, Pedro Luiz Serrano Uson Junior, Geoffrey Kannan, Haiyong Han, Daniel D. Von Hoff

**Affiliations:** 1Department of Oncology and Hematology, Emory University, Atlanta, GA 30322, USA; 2Drug Development, HonorHealth Research and Innovation Institute, Scottsdale, AZ 85258, USA; 3Center for Personalized Medicine, Hospital Israelita Albert Einstein, São Paulo 05652-900, Brazil; 4Drug Development, LabCorp, Princeton, NJ 08540, USA; 5Basic Research Unit, Pancreatic Cancer Program, Translational Genomics Research Institute (TGen), Phoenix, AZ 85004, USA; 6Molecular Medicine Division, Translational Genomics Research Institute (TGen), Phoenix, AZ 85004, USA

**Keywords:** unfolded protein response (UPR), endoplasmic reticulum (ER) stress, pancreatic cancer, apoptosis

## Abstract

High rates of cell proliferation and protein synthesis in pancreatic cancer are among many factors leading to endoplasmic reticulum (ER) stress. To restore cellular homeostasis, the unfolded protein response (UPR) activates as an adaptive mechanism through either the IRE1α, PERK, or ATF6 pathways to reduce the translational load and process unfolded proteins, thus enabling tumor cells to proliferate. Under severe and prolonged ER stress, however, the UPR may promote adaptation, senescence, or apoptosis under these same pathways if homeostasis is not restored. In this review, we present evidence that high levels of ER stress and UPR activation are present in pancreatic cancer. We detail the mechanisms by which compounds activate one or many of the three arms of the UPR and effectuate downstream apoptosis and examine available data on the pre-clinical and clinical-phase ER stress inducers with the potential for anti-tumor efficacy in pancreatic cancer. Finally, we hypothesize a potential new approach to targeting pancreatic cancer by increasing levels of ER stress and UPR activation to incite apoptotic cell death.

## 1. Introduction

Pancreatic cancer continues to be the third most common cancer-related death in the United States, accounting for 7% of all cancer deaths and 3% of all cancers [1]. Pancreatic ductal adenocarcinoma (PDAC) develops in the exocrine pancreas and is by far the most prevalent pancreatic cancer, accounting for more than 90% of all cases [2]. As of 2022, it is estimated that 62,210 people will be diagnosed with pancreatic cancer and 49,830 will die of the disease, with a current five-year survival rate of 11% [1]. Proportionally, men are at a slightly higher risk than women for pancreatic cancer.

Precision medicine with the incorporation of universal sequencing may improve the care of patients. However, only around one in six patients harbor a pathogenic germline variant, among which around half are in homologous recombination repair-related genes [3]. In patients with pathogenic BRCA1/2 or PALB2 mutations, the incorporation of platinum-based therapies and PARP inhibitors such as Olaparib brings modest improvements in outcomes [3,4,5]. More recently, two patients were treated with TCR gene therapy targeting the KRAS G12D driver mutation. One patient with metastatic refractory pancreatic cancer had a response after a single infusion of autologous T cells, which was continued at 6 months [6].

Poor prognosis in pancreatic cancer could be related to primary chemo-resistance [7]. Desmoplasia and the tumor microenvironment are two factors implicated in resistance to treatment, such as primary resistance to immunotherapy. Understanding and targeting cellular homeostasis could help to improve drug delivery and ultimately treatment outcomes [8]. Several experimental studies examined the unfolded protein response (UPR) and its correlation with cellular homeostasis and stress as a possible target in cancer [9]. As pancreatic cancer cells are severely hypoxic and bear cell stress, we hypothesize that increasing cell stress might induce apoptosis in pancreatic tumor cells [10]. In this article, we will review and examine the mechanisms involved in cell and ER stress, the role of the UPR in pancreatic cancer and its activation, and strategies to increase cell stress and tumor cell death through UPR activation.

## 2. Mechanisms of Cellular Homeostasis

In cellular metabolism, the mitochondria are involved in several essential functions such as energy generation, calcium signaling, stress response, cell differentiation, and apoptosis [11,12,13]. In the case of defective mitochondria, cellular homeostasis may be compromised, and quality control mechanisms are activated to preserve the balance between homeostasis and cell death [14]. Mitophagy can also be triggered through multiple signaling cascades in response to persistent defects [14].

An important part of cellular homeostasis with relevance to cancer metabolism is protein homeostasis. The endoplasmic reticulum (ER) is an interconnected single membrane-bound network that serves as a protein factory, where about one-third of all proteins, secretory or membrane-associated, are folded [15]. An ER quality control system exists to preserve the balance of the folded proteins [16]. This control system can detect correctly folded proteins exiting the ER to their destination, as well as misfolded or unfolded proteins [15]. This mechanism prevents misfolded or unfolded proteins from transiting the secretory pathway and assures that misfolded proteins are directed toward a degradative pathway [16,17]. ER-associated degradation (ERAD) is the process by which proteins that are terminally misfolded are transported from the ER to the cytosol and on to proteasome degradation [15].

To correctly recognize and promote the ubiquitination of misfolded proteins, the ERAD system is composed of multiple components that integrate protein complexes with the ER membrane [15]. In yeast, three membrane protein complexes that define different complexes (ERAD-L, ERAD-M, and ERAD-C) have been identified and proposed [15]. When proteins are misfolded, it is proposed that they are degraded by these complexes [15]. The progressive accumulation of unfolded proteins activates the ER stress receptors, which, together with molecular chaperones, activate the ERAD system and the unfolded protein response (UPR) to enhance the clearance of misfolded proteins [15].

The UPR is an adaptive response to the activation of the Integrated Stress Response (ISR) [17]. The ISR is an evolutionarily conserved intracellular signaling network that is activated when a cell experiences various intrinsic or extrinsic insults. Extrinsic activators of the ISR include amino acid depletion, glucose deprivation, and hypoxia, among others [18]. Conversely, there are intrinsic insults to the cell and, in particular, the ER can activate the ISR and specifically the UPR to induce changes in intraluminal calcium, altered glycosylation, nutrient deprivation, pathogen infection, expression of folding-defective proteins, and changes in redox status [17]. UPR activation may lead to resolved homeostasis, but under prolonged and unresolved stress, the signaling pathway will progress to apoptotic mechanisms [17].

## 3. Mechanisms of Unfolded Protein Response and Cancer

Beyond the multiple functions of the UPR in cellular homeostasis, cancer development, and maintenance, of particular interest are the endoplasmic reticulum (ER) membrane stress sensors [17]. These stress sensors include inositol-requiring enzyme 1 (IRE1), double-stranded RNA-activated protein kinase (PKR)–like ER kinase (PERK), and activating transcription factor 6 (ATF6) [17,18,19,20,21,22]. PERK is one of the four transmembrane initiators of intracellular signaling that comprise the ISR. Increasing evidence shows that UPR sensors are related to oncogenic programming and cancer stem cells (CSC), including the work by Pattabiraman and Weinberg, as well as the work of Ingrid Caras, which have propelled further research and clinical trials into CSC-targeted therapies [23,24].

As the ER maintains the correct balance of secreted and transmembrane proteins, the UPR functions as an adaptive response to the accumulation of folded proteins and the imbalances of the capacity of ER protein folding and demand [17,25,26,27]. The ER transmembrane sensors IRE1, PERK, and ATF6 regulate proteostasis and homeostasis through separate and overlapping pathways [16] (Figure 1).

In cancer, the UPR plays a role in cell survival properties, carcinogenesis, and even mechanisms related to treatment resistance [17,28,29]. As a result of aberrant protein production and intrinsic stress in cancer, the UPR is highly active [17]. The tumor microenvironment is normally subjected to stress by intrinsic and extrinsic conditions (Table 1), including nutrient shortage, hypoxia, aberrant protein synthesis, and chemotherapy exposure [30,31,32]. These conditions can also upregulate UPR activation, with cellular homeostatic effects and consequently tumor survival. Furthermore, it is shown that the UPR is related to multiple other factors in cancer, including genomic instability, inflammation, cell survival, angiogenesis, and metastatic properties [30].

Recent efforts have sought to better characterize the role of inflammation in tumorigenesis, specifically regarding the implication of chronic pancreatitis (CP) in pancreatic cancer development. Lowenfels et al. [33] and several meta-analyses have confirmed that CP patients face an increased risk of developing PDAC at the 10- and 20-year mark [34,35,36]. Although many factors may contribute to the development of PDAC from CP, one leading hypothesis is that chronic inflammation presents an optimal environment for oncogenic mutations, which could catalyze carcinogenesis for patients with a genetic predisposition or at environmental risk (tobacco, nicotine, or alcohol abuse) for pancreatic cancer [37]. Additionally, with the advancement of genetically engineered mouse models, the identification of acinar cells as the main cellular origin of PDAC and the transdifferentiation of acinar cells into pancreatic intraepithelial neoplasia (PanIN) adds context to the link between PDAC and CP [38].

Acinar-to-ductal metaplasia (ADM) is the process of pancreatic acinar cells differentiating into ductal-like cells [38]. Because acinar cells are synthetically very active and prone to ER stress, their plasticity represents an intrinsic defense mechanism to protect acinar cells from genetic and environmental pressures [38]. However, under sustained cell stress, such as inflammation during chronic pancreatitis and mutant KRAS or aberrant growth factor signaling, acinar cells differentiate through ADM into PanIN, a precancerous lesion and the main pathological basis of PDAC development [38]. Hingorani et al. determined the origin of PDAC using KRAS^G12D^ mice and found that PanIN formation was preceded by acinar-to-ductal metaplasia [39]. Furthermore, Carrière et al. subjected genetically engineered mice with a Lox-Stop-Lox (LSL) sequence followed by a K-Ras G12D point mutation (LSL-KRAS^G12D^ mice) to a series of caerulein injections (previously demonstrated to induce pancreatitis) and found that pancreatitis-induced acinar cell regeneration increased the propensity for pancreatic malignant transformation, confirming the role of inflammation in the development of PanINs and subsequent PDAC [40].

## 4. Evidence for UPR Being Important in Pancreatic Cancer

The UPR could be a pathway with therapeutic potential in PDAC [32]. In a study evaluating the cytotoxic effect of nemorosone, a polycyclic polyprenylated acylphloroglucinol derived from an ethanolic flower extract of *Clusia Rosea* on pancreatic cell lines, UPR response was induced after ER stress [32]. Pancreatic cell lines Capan-1, AsPC-1, and MIA-PaCa-2 were exposed to nemorosone in increasing concentrations. At 10 mM, a significant reduction in pancreatic cancer cell growth of at least 80% was observed. Among multiple effects observed, an evaluation of the expressed genes on the tested lines revealed them to be significantly associated with the response to cellular stress, as well as the regulation of apoptosis and cell cycle. Furthermore, most induced genes shared by all cell lines were found to be directly involved in the signaling network of ER stress and the UPR [32]. Many of the genes are induced by ATF4 and resemble pro-survival genes related to the UPR [32].

There is growing evidence that ER stress and UPR activation are abundant in pancreatic cancer [7,41]. In 2015, Kong et al. examined KRAS/Mek-mTOR signaling by implanting pancreatic tissue from PDAC patients who had pancreatic resections into wild-type mice, thus creating transgenic p48^Cre/+^, LSL-KRAS^G12D/+^, and Tsc1^flax/+^ murine models [41]. By establishing these cell lines and implanting them orthotopically into wild-type mice, researchers found extensive necrotic components and confirmed high levels of hypoxia-induced ER stress in the tumor microenvironment [41]. A particular UPR component of interest is the ER chaperone protein glucose-related protein 78 (GRP78) [42,43]. GRP78, also called BiP (immunoglobulin heavy chain-binding protein) or HSPA5 (heat-shock protein A5), binds misfolded and unfolded proteins to correct the folding process. Additionally, it is upregulated in the context of ER stress, along with PERK and the other complexes that reduce protein synthesis [43,44]. GRP78 blocks the activation of IRE1, PERK, and ATF6 by binding to their ER luminal domains. Under ER stress, GRP78 dissociates, resulting in the activation of IRE1, PERK, and ATF6 and their downstream signaling pathways (Figure 1).

Elevated GRP78 is related to poor prognosis in pancreatic cancer [45]. In one study, GRP78 was detected using tissue microarray-based immunohistochemistry in tissues from 180 pancreatic cancer patients [44]. Interestingly, not only was worse overall survival and a higher T-stage observed among patients with higher expression of GRP78 but the expression of GRP78 was also higher in tumor tissues than in the adjacent nontumor tissues [44]. In vitro, the regulation of GRP78 in PDAC cell lines affected the proliferation, migration, and invasion capacities [45]. Transcriptomic analysis of the PDAC cell line S2-VP10 evaluating the shRNA-mediated knockdown of GRP78 suggests that GRP78 interferes in multiple signaling pathways including cell-cycle, apoptosis, and actin-cytoskeleton regulation [45]. In Pdx1^-Cre^, Kras^G12D/+^, and p53^f/+^ murine models of PDAC, high GRP78 was also related to tumor development; however, in the same genetic background, mice pancreatic tissue bearing an additional GRP78^f/+^ allele reduced pancreatic tumorigenesis [46]. Resistance to chemotherapy can also be related to GRP78. Silencing GRP78 reduced efflux by ATP-binding cassette transporters, and in combination with chemotherapy, cells treated with silencing GRP78 exhibited significantly more cell death [47].

The relationship between IRE1 and the proliferative effects in pancreatic cancer cell lines has been reported [48]. Proliferation assays using fourteen pancreatic cancer cell lines showed a dose- and time-dependent growth inhibition by IRE1α-specific inhibitors, and subsequent cell cycle analysis showed that these IRE1α inhibitors caused growth arrest at either the G1 or G2/M phases (SU8686, MiaPaCa2) and induced apoptosis (Panc0327, Panc0403) [48]. In another study, suppression of IRE1 with IRE inhibitor STF-083010 alone reduced the viability of pancreatic cancer cell lines [49]. In the same study, sunitinib-, gemcitabine-, and chloroquine-treated mice showed a significant reduction in GRP78 expression, reduced cell proliferation, and increased apoptosis [48]. Studies in pancreatic cancer cell lines also showed that PERK activation can contribute to tumorigenesis [48]. GSK2656157, an ATP-competitive inhibitor, showed inhibition of stress-induced PERK autophosphorylation, eIF2α substrate phosphorylation, and decreased ATF4 and CAAT/enhancer binding protein–homologous protein in multiple cell lines [49]. Further oral administration of GSK2656157 resulted in dose-dependent inhibition of multiple human tumor xenograft growth in mice [49]. HSP70, similar to GRP78, is another chaperone that can trigger UPR-mediated cell death [50]. Modulating HSP70 is also another strategy to induce the UPR and stress, and agonists and antagonists are being developed as therapeutic strategies to modulate the UPR [50].

## 5. Molecular Induction of ER Stress and Apoptosis

During tumorigenesis, rapid cell proliferation and high rates of protein synthesis deplete the tumor microenvironment (TME) of oxygen, nutrients, and glucose, leading to ER stress [7]. In addition, the heavy demand and limited capacity of ER protein folding during cell proliferation lead to an accumulation of improperly folded proteins. Under general ER stress such as abnormal TME, IRE1α, PERK, and ATF6 have unique mechanisms to promote cell survival and regulate cellular homeostasis. However, failure to restore cellular homeostasis under prolonged ER stress may lead to adaptation, senescence, or cell death through apoptosis, as depicted in Figure 2.

Mitigating ER stress and the effects of the UPR have been proposed for several cancers and diseases, particularly hematologic malignancies, where HSP90 inhibitors reduce ER stress by inhibiting the post-translational modification of proteins, thus reducing protein levels vital for cancer pathology and disrupting signaling cascades. Though many theories propose reducing ER stress, we propose a radically different approach of mechanistically increasing ER stress to levels the cell cannot overcome to induce apoptosis through the UPR, thus enhancing the effects of chemotherapy. A summary of agents and treatment modalities under recent investigation for their ability to induce ER stress and apoptotic cell death are listed below in Table 2.

## 6. Gambogenic Acid (GNA)

Gambogenic acid (GNA), a compound isolated from the traditional Chinese medicine gamboge, has shown promise in inducing apoptosis by increasing ER stress through several different mechanisms. In colorectal cancer (CRC) xenografts and murine models, GNA induced ER stress-mediated apoptosis in a reactive oxygen species (ROS)-dependent manner, activating the IRE1α pathway to induce prolonged ER stress [51]. More recently, GNA was shown to inhibit CRC proliferation in vitro by inducing apoptosis [52]. Upon further examination, while GNA upregulated the expression of IRE1α, PERK, and downstream eIF2α and ATF4, GNA-induced ER stress was deduced to occur from inhibition of protein synthesis resulting from eIF2α phosphorylation [52]. When GNA’s effect was examined in vivo using an azoxymethane (AOM)/dextran sulfate sodium (DSS) mouse model of colitis-associated cancer (CAC) established in BALB/c mice, GNA clearly suppressed colorectal tumors while also increasing the expression of CHOP and GRP78, indicating GNA suppressed tumor growth by triggering ER stress [52]. In nasopharyngeal carcinoma, however, in addition to increasing GRP78 and CHOP expression, treatment with GNA resulted in a dramatic efflux of intracellular chloride ions and an opening of the chloride ion channel [53]. As apoptotic volume decrease (AVD) through chloride channels is an early hallmark of apoptotic events, this suggests a possible pleiotropic mechanism in which GNA induces apoptosis through both ER stress and chloride channel opening.

## 7. Tigatuzumab

Aside from increasing ER stress, immunotherapies known to induce apoptosis through caspase activation by targeting proteins associated with mitochondrial are also of interest. Tigatuzumab, a human monoclonal antibody that acts as a death receptor 5 agonist, has shown preclinical promise in inducing tumor necrosis factor-related apoptosis-inducing ligand (TRAIL) activity in tumor cells [54]. TRAIL, a member of the TNF superfamily of cytokines, is expressed commonly in colon, gastric, pancreatic, and other types of human cancers, with little to no expression in normal tissue and has a marked ability to bind death receptor 5 (DR5) and trigger apoptosis in tumor cells via downstream caspase activation [55]. As TRAIL targets multiple receptors, agonistic monoclonal antibodies such as tigatuzumab that specifically bind DR5 may be advantageous anticancer agents due to a more targeted and stronger apoptosis-inducing activity [54]. Preclinical studies of TRA-8, a murine DR5 agonist, in pancreatic cancer cell lines showed single-agent anti-tumor efficacy and increased efficacy in combination with doxorubicin, paclitaxel, and gemcitabine [56]. Increased sensitivity to treatment was observed in tissue with higher DR5 expression levels [56]. The clinical efficacy of Tigatuzumab in combination with FDA-approved chemotherapeutic agents is described in Table 3.

## 8. Minnelide

Preclinical studies demonstrated that triptolide, a diterpenoid triepoxide, had high efficacy in inhibiting pancreatic cancer cells in vitro and blocking tumor growth and metastasis in vivo [59]. Mujumdar et al. unveiled how triptolide induces cell death by increasing ER stress and activating UPR-mediated apoptosis [59]. Mechanistically, triptolide initially increased GRP78 expression 4 h after treatment, reflecting an initial cellular survival mechanism, but decreased GRP78 expression by the 24 h timepoint, activating the ER stress pathway to initiate apoptotic death [59]. Moreover, treatment with triptolide resulted in sustained increases of IRE1α and eIF2α levels, induced the expression of CHOP mRNA in multiple cell lines, and led to increased levels of ER stress, indicating that treatment with triptolide induces the UPR in pancreatic cancer cells through the PERK-eIF2α and IRE1α pathways [59].

Given triptolides’ poor solubility limiting its clinical use, Minnelide was developed as a highly water-soluble analog that exhibited a greater preclinical efficacy, showing great potential for its clinical application. In a Phase 1 dose escalation and pharmacokinetic study of Minnelide in pancreatic and gastric tumors, 36% of patients saw a partial metabolic response following cycle 1, and stable metabolic disease was observed in 52% of patients (*n* = 19) [60]. Using RECIST criteria (*n* = 10) after cycle 2, a partial response was observed in one patient (gastric) and stable disease was seen in six patients (5 pancreas, 1 rectal). Although neutropenia was observed as a common dose-limiting toxicity, this was rapidly reversible, and a revised treatment schedule has been developed to allow for more sustained treatment [60]. Currently, a Phase 1 clinical trial is underway, evaluating the treatment-related adverse events and anti-tumor activity of Minnelide alone and in combination with paclitaxel in patients with advanced solid tumors, mainly pancreatic and gastric (NCT03129139).

## 9. BOLD-100/KP1339

BOLD-100/KP1339, a ruthenium (III)-based anticancer agent, has shown promising anticancer activity in preclinical studies on solid tumors for its ability to increase ER stress [61]. Compared to platinum-based anticancer agents that target DNA, this ruthenium-based metallic agent acts on proteins and has been shown to increase the production of ROS while downregulating GRP78 and upregulating PERK and downstream CHOP, increasing ER stress and promoting ER-dependent apoptosis [61,62]. In a Phase I clinical trial, stable disease was observed for patients with non-small-cell lung cancer, sarcoma, and colorectal cancer tumors, and a partial response or disease stabilization was observed in patients with gastrointestinal neuroendocrine tumors [63]. Given the documented safety with manageable adverse effects observed in the Phase I clinical trial, a Phase 1B trial is currently underway investigating the combination of BOLD-100/KP1339 and FOLFOX for patients with advanced solid tumors (NCT04421820).

## 10. Radiation Therapy to Activate UPR

Ionizing radiation is known to generate ROS and reactive nitrogen species causing oxidative damage to macromolecules in the mitochondria, leading to ER stress and activation of the UPR. With a multitude of clinical studies suggesting only around a 30% response rate for pancreatic cancer to radiation therapy [65,66,67,68,69,70], a combination with ER stress inducers may offer clinical benefits for overcoming radioresistance [70]. In glioblastoma multiforme (GBM), when exposing patient-derived GBM stem cells (GSCs) to the ER stress-inducing agent 2-deoxy-D-glucose (2-DG), dose-dependent decreases in tumor viability were observed, along with increased apoptotic markers, indicating that increasing ER stress pharmacologically sensitized GBM cells to radiotherapy [70]. Although the pharmacological induction of ER stress has yet to be investigated in combination with radiation therapy in pancreatic cancer, activating the CHOP-mediated apoptosis pathway through ER stress-inducing agents may potentially sensitize pancreatic cancer tumors to radiation and improve response rates.

## 11. Conclusions

Given the heterogeneity of pancreatic cancer in terms of mutations, stressors, and tumor microenvironment by the time of detection, identifying a common context of vulnerability expressed across a wide array of cases could be an attractive approach to broadly targeting this aggressive malignancy. Several preclinical and clinical investigations over the past decade have presented evidence of both intrinsic and extrinsic stressors in the tumor biology of pancreatic cancer, contributing to increased ER stress and the activation of the unfolded protein response. Although UPR activation canonically serves to enable tumors to proliferate under mild to moderate ER stress, we hypothesize that increasing the load and prolonging ER stress may be an Achilles’ heel in this adaptive mechanism to stress cells to induce apoptotic cell death. Here, we present several therapeutics with the potential to increase ER stress and critically enhance the effects of currently used chemotherapies. Further preclinical models and clinical phase investigations are warranted to better characterize the safety and efficacy of these therapeutics and offer insight into the potential anti-tumor effects of ER stress inducers. In particular, the downstream effectors of apoptosis need to be well understood, along with the stress conditions that potentiate their activation. This information will lead to more refined therapeutics. Further clinical-stage research will also examine the toxicities in patients associated with modulating ER stress, which will be critical information for developing therapeutics that exploit the UPR pathway.

## Figures and Tables

**Figure 1 ijms-24-00577-f001:**
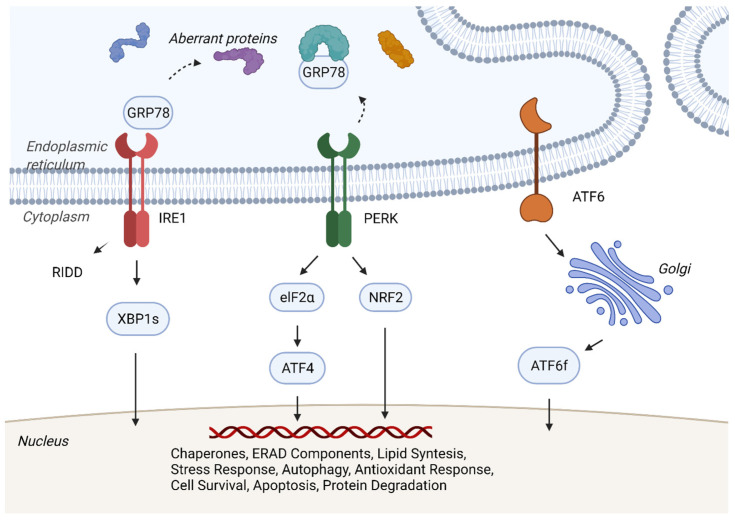
Unfolded Protein Response. A simplified version of unfolded protein response (UPR). Under stress, cells activate the adaptative UPR to establish cellular homeostasis and proteostasis. The sensors IRE1, PERK, and ATF6 are activated after the dissociation of GRP78, which is recruited due to the accumulation of misfolded and unfolded proteins. IRE1 oligomerization and auto-phosphorylation induce kinase and RNase activity leading to mRNA degradation (RIDD) and altered splicing of the XBP1 mRNA and production of the transcription factor XBP1s. PERK undergoes oligomerization and auto-phosphorylation, subsequently phosphorylating the eukaryotic translation initiation factor eIF2α, resulting in the translation of transcription factor ATF4. NRF2 is also phosphorylated by PERK and activates antioxidant responses. ATF6 activation occurs after transport to the Golgi apparatus where it undergoes S1P/S2P protease cleavage, yielding transcriptionally active ATF6f. The transcriptional events mediated by XBP1s, ATF4, and ATF6f promote the expression of chaperones and components of ER-associated protein degradation (ERAD), reducing protein translation and increasing ER capacity in order to restore homeostasis. Additionally, UPR-mediated gene expression also directly impacts autophagy, cytokine production, and apoptosis. Figure 1 was created with BioRender.com.

**Figure 2 ijms-24-00577-f002:**
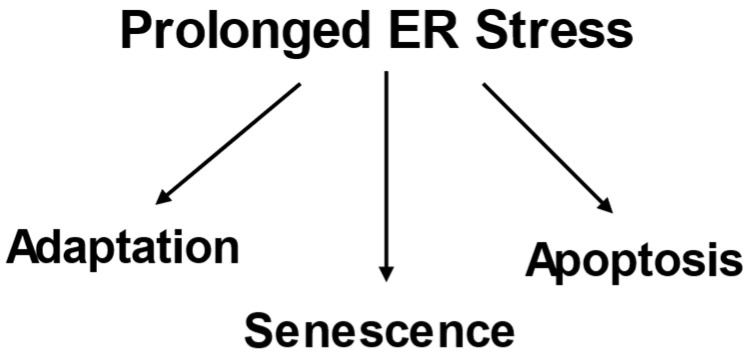
Cellular Responses to Prolonged Levels of ER Stress.

**Table 1 ijms-24-00577-t001:** Factors that increase stress.

Intrinsic Factors	Extrinsic
Oncogenic activation	Hypoxia
Altered Ploidy	Acidosis
Exacerbated Secretors	Nutrient depletion
Genomic Instability	VEGF
Redox imbalance	
Inward mutation	

**Table 2 ijms-24-00577-t002:** Potential Agents to Activate UPR-Induced Apoptosis in Pancreatic Cancer.

Agent	Mechanism of Action	Stage of Development	Clinical Trials	Comments
Gambogenic Acid (GNA)	ROS-dependent activation of IRE1α leads to prolonged ER stress; PERK activation → eIF2α phosphorylation inhibits protein synthesis [51,52,53]	Preclinical		No first-in-human trials
Tigatuzumab	Death Receptor (DR)5 monoclonal antibody agonist; Induces TRAIL to bind DR5 initiating downstream caspase activation in tumors [54,55,56,57,58]	Phase 2	NCT01307891 NCT01220999	Combination with gemcitabine, sorafenib, nab-PAC
Minnelide	Water-soluble analog of triptolide; Inhibits GRP78; Upregulates IRE1α and PERK pathways to increase ER stress [59,60]	Phase 1/2	NCT03129139 NCT04896073	Single-agent and combination trials with Paclitaxel
BOLD-100	Ruthenium (III) anticancer agent; inhibits GRP78 and increases ROS production; leads to ER stress [61,62,63]	Phase 1B	NCT04421820	Combination with FOLFOX
Disulfram (Antabuse)	Binds Copper to form DSF-Cu complex; Induces ROS production → Increased levels of oxidized proteins → ER stress and UPR activation [64]	Phase 2 Completed, Phase 1 (recruiting)	NCT03714555 NCT02671890	Combination with nab-PAC-gemcitabine, FOLFIRINOX, Gemcitabine
Radiation therapy	Radiation → ROS/RNS production → ER stress induction and UPR activation [65,66,67,68,69,70]			ER stress inducers sensitize cancer cells to radiation treatment
KRN5500 (Spicamycin analog)	Anti-Golgi drug; Inhibits protein synthesis and glycoprotein processing via altered Golgi (dilated cisternae) → Accumulation of unfolded proteins in ER lumen → apoptosis via intrinsic pathway [71,72]	Phase 1 completed	NCT00017238 NCT00002923	No tumor response, three disease stabilizations observed in in-human trials.
Atovaquone	Ubiquinone analogue: Inhibits Complex III of ETC and oxidative phosphorylation → leads to oxidative and ER stress [73,74]	Phase 1	NCT04648033	Investigated in NSCLC, ovarian cancer
Auranofin	Gold (I) complex; inhibits thioredoxin reductases (TrxRs) → increased ROS species leading to ER stress; also targets PI3K/AKT/mTOR pathway [75,76]	Phase 1	NCT01747798 NCT03456700	Investigated in NSCLC, ovarian cancer

Footnote: nab-PAC: nab-paclitaxel, CRC: Colorectal cancer NSCLC: Non-Small Cell Lung Cancer, FOLFIRINOX: fluorouracil, folinic acid, irinotecan, oxaliplatin, FOLFOX: fluorouracil, oxaliplatin + folinic acid.

**Table 3 ijms-24-00577-t003:** Phase 2 clinical efficacy of Tigatuzumab (TIG) in combination with gemcitabine, sorafenib, and nab-PAC/Abraxane.

Trial Phase	Combination Agents	Type of Cancer	Progression-Free Survival (PFS)	Overall Survival (OS)	Conclusions
Phase II	Tigatuzumab + Gemcitabine	Unresectable or metastatic pancreatic cancer	52.5% PFS at 16 weeks; not significant from historical data at 44% seen with gemcitabine alone [57]	8.2 months; comparable to 3.6–6.8 months (gemcitabine alone), 3.8–11.1 months (gemcitabine + other agents), 11.1 months (FOLFIRINOX trial) [57]	Marginal increase in overall survival with TIG compared to gemcitabine alone suggests possible contribution of TIG to anti-tumor effects of gemcitabine; TIG may be clinically active [57]
Phase II	Tigatuzumab + Sorafenib	Advanced hepatocellular carcinoma	Time to progression (TTP): 3.9 months in 6/6 mg/kg TIG + SOR; 2.8 months in SOR alone (small sample size *p* = 0.988) [54]	12.2 months in TIG + SOR; 8.2 months in SOR alone (small sample size *p* = 0.737) [54]	TIG + SOR failed to meet primary efficacy endpoint of TTP. However, combination was well tolerated and suggests possible increase in OS for TIG [54]
Phase II	Tigatuzumab + nab-PAC/Abraxane	Metastatic triple-negative breast cancer (TNBC)	2.8 months overall in TIG + nab-PAC, 3.8 months in patients with objective response; 3.7 months in nab-PAC arm [58]	Overall response rate (ORR): 28% (CI 14.9–45.0% in TIG + nab-PAC; 38% (CI 18.0–61.1%) in nab-PAC arm [58]	3 complete responses (CR) + 1 near CR in TIG + nab-PAC arm; no CR in nab-PAC arm; does not support further research of TIG + nab-PAC; however, notable increase in complete responses suggests further investigation of anti-DR5 agents [58]

## Data Availability

Not applicable.

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
