# Peer review of "Increasing Stress to Induce Apoptosis in Pancreatic Cancer via the Unfolded Protein Response (UPR)"

_ijms, 2022, doi:10.3390/ijms24010577_

Round 1

Reviewer 1 Report

This is a very good review, which can be published with appropriate language embellishment.

Reviewer 2 Report

A review from Botrus et al. talks about the role of endoplasmic reticulum stress and unfolded protein response on pancreatic cancer cell apoptosis. The review is nicely written and can be elaborated to attract a broad audience by incorporating the following points.

1.    Pancreatic acinar cells are synthetically very active and are prone to ER stress. Authors should also include a section explaining the role of inflammation (pancreatitis) induced stress in chronic pancreatitis and its development into pancreatic cancer.

2.    The authors should also describe in little detail the genetically engineered mouse models used to analyze ER stress in pancreatic cancer and pancreatitis.

3.    Multiple examples or studies should explain the vital or recent information wherever discussed in the text.
